# Simple and Effective Stochastic Neural Networks

## Abstract

Stochastic neural networks (SNNs) are currently topical, with several paradigms being actively investigated including dropout, Bayesian neural networks, variational information bottleneck (VIB) and noise regularized learning. These neural network variants impact several major considerations, including generalization, network compression, and robustness against adversarial attack and label noise. However, many existing networks are complicated and expensive to train, and/or only address one or two of these practical considerations. In this paper we propose a simple and effective stochastic neural network (SE-SNN) architecture for discriminative learning by directly modeling activation uncertainty and encouraging high activation variability. Compared to existing SNNs, our SE-SNN is simpler to implement and faster to train, and produces state of the art results on network compression by pruning, adversarial defense and learning with label noise.

## 1 Introduction

Stochastic neural networks (SNNs) have a long history. Recently various stochastic neural network instantiations have been topical in their applications to reducing overfitting (Gal & Ghahramani, 2016; Neelakantan et al., 2015) and training data requirements (Garnelo et al., 2018), providing confidence estimates on predictions (Gal & Ghahramani, 2016), enabling network compression (Dai et al., 2018), improving robustness to adversarial attack (Alemi et al., 2017), improving optimization (Neelakantan et al., 2015), generative modeling (Kingma & Welling, 2014), and inputting or producing probability distributions (de Bie et al., 2019; Frogner et al., 2019).

One of the most theoretically appealing stochastic neural network formulations is Bayesian neural networks, which place a prior distribution on the weights of the network (Graves, 2011; Blundell et al., 2015; Ritter et al., 2018). However this usually necessitates more complex learning and inference procedures that rely on variational approximations or sampling. Another group of works instead focus on modeling the uncertainty in neural network activations. Notably the variational information bottleneck (VIB) approach (Alemi et al., 2017) is motivated by information theoretic considerations (Tishby et al., 1999) to learn a hidden representation that carries maximum information about the output and minimum information about the input. Evaluating the required mutual information terms requires modeling probability distributions over activations rather than weights. Deep VIB (Alemi et al., 2017) leads to improved generalization, adversarial robustness and model compression algorithms (Dai et al., 2018). Furthermore, modeling stochastic activations through noise is often practically useful for improving exploration and local minima escape during optimization, generalization and adversarial robustness (You et al., 2018; Noh et al., 2017; Bishop, 1995; Gulcehre et al., 2016), and in some cases can be linked back to Bayesian models of weights (Noh et al., 2017; Gal & Ghahramani, 2016) when the noise added at each activation can be considered as a result of different samples from the weight posterior.

In this paper we propose a simple and effective stochastic neural network (SE-SNN) that models activation uncertainty through predicting a Gaussian mean and variance at each layer, which is then sampled during the forward pass. This is similar to the strategy used to model activation distributions in VAE (Kingma & Welling, 2014) and VIB (Alemi et al., 2017). Differently, we then place a non-informative prior on activation distribution and derive an activation regularizer that directly encourages high activation variability via preferring high-entropy activations. In conjunction with a discriminative learning loss, this means that the network is optimized for activation patterns that have

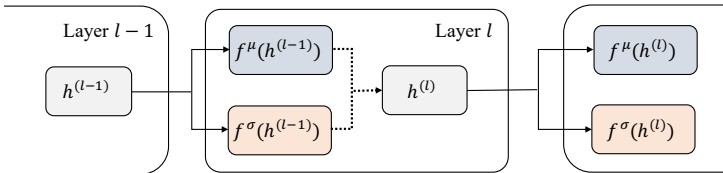

Figure 1: An illustration of the stochastic learning module in a SE-SNN. The output of layer $l$ is sampled from the learned distribution defined by $f^\mu(h^{l-1})$ and $f^\sigma(h^{l-1})$.

high uncertainty while simultaneously being predictive of the target variable. The interplay between these two objectives leads to several appealing capabilities in pruning, adversarial defense and learning with label noise. **Pruning**: Optimizing for high per-activation variability/uncertainty and predictive accuracy simultaneously lead to the network packing more entropy into the least significant neurons – so that the most crucial neurons are free to operate unperturbed. This leads to a simple pruning criterion based on each neuron's entropy value. **Adversarial defense**: By optimizing for both per-activation uncertainty and the network's predictive accuracy, a representation-level data augmentation policy is trained that perturbs the internal features during training for increased robustness (Alemi et al., 2017; You et al., 2018). **Label noise**: With SE-SNN, per-activation uncertainty can be easily aggregated to produce per-instance uncertainty. By optimizing for per-instance uncertainty and predictive accuracy, the network allocates the uncertainty to spread the representation and prediction of the hard-to-classify instances so as to downweight their influence on parameter learning. The result is a model robust to label noise as well as outlying training samples.

To summarize, our contributions are: (1) A new simple yet effective stochastic neural network formulation. (2) We show that our SE-SNN has connections to VIB (Alemi et al., 2017), Dropout (Srivastava et al., 2014) and non-informative activation priors while being simpler to implement and faster to train, as well as impactful on a variety of practical problems. (3) Comprehensive evaluations show excellent performance on pruning-based model compression, adversarial defense, and label noise robust learning.

## 2 METHODOLOGY

**Stochastic Layers**    We consider a neural network discriminatively trained for a predictive task such as object recognition. Instead computing fixed point estimates of feature vectors, we propose to use *stochastic* neurons. More specifically, for an input $h$, a layer will output a series of univariate distributions. By sampling from those distributions independently, we get a random output $z$. Finally we apply the non-linear activation function $\psi(\cdot)$ to $z$ and get the input for the next layer. In this study, we choose to use Gaussian distribution with parameterized mean and variance, which has been popularized by VAE (Kingma & Welling, 2014) and VIB (Alemi et al., 2017) due to the ease of reparameterization. Formally, for the $l$-th layer, this forward-pass process can be written as (omitting neuron index for notation simplicity),

$$
\begin{aligned}
\mu^{(l)} &= f^\mu(h^{(l-1)}), \\
\sigma^{(l)} &= f^\sigma(h^{(l-1)}), \\
z^{(l)} &\sim \mathcal{N}(\mu^{(l)}, \sigma^{(l)}), \\
h^{(l)} &= \psi(z^{(l)}).
\end{aligned}
\tag{1}
$$

This process is illustrated in Figure 1, where each standard deviation predictor $f^\sigma$ comes with a softplus activation $f(x) = \log(1 + \exp(x))$ to ensure non-negativity.

**Supervised Learning Loss**    We can choose to replace some or all intermediate layers of a vanilla neural network with such stochastic layers. For the final layer (i.e., the classifier) we opt for a standard linear layer, and the classification loss is the same as that of a vanilla neural network, e.g., cross-entropy. Since a Gaussian distribution is fully reparameterizable, the network can be trained end-to-end as long as the sampling process $z \sim \mathcal{N}(\mu, \sigma)$ is realized by $z = \mu + \epsilon \cdot \sigma$ where $\epsilon \sim \mathcal{N}(0, 1)$.

**Max-entropy Regularization**  We place a non-informative prior on the produced Gaussian (denoted as $\mathcal{N}(\mu_1, \sigma_1)$). The non-informative prior is a Gaussian with arbitrary mean ($\mu_1$) and infinite variance ($\sigma_1^2$). This reflects the prior that none of neurons is meaningful for predictive purposes. The non-informative prior leads to a regularization term that minimizes the KL divergence of the produced Gaussian ( $\mathcal{N}(\mu_2, \sigma_2)$) and the infinite-variance Gaussian

$$\min_{\mu_1, \sigma_1} (\lim_{\sigma_2 \to \infty} \mathrm{KL}(\mathcal{N}(\mu_1, \sigma_1) || \mathcal{N}(\mu_2, \sigma_2))), \qquad \forall \mu_2 \in \mathbb{R}$$

$$\Rightarrow \min_{\mu_1, \sigma_1} (\lim_{\sigma_2 \to \infty} (\log \frac{\sigma_2}{\sigma_1} + \frac{\sigma_1^2 + (\mu_1 - \mu_2)^2}{2\sigma_2^2} - \frac{1}{2})), \qquad \forall \mu_2 \in \mathbb{R}$$

$$\Rightarrow \min_{\sigma_1} (\lim_{\sigma_2 \to \infty} (\log \frac{\sigma_2}{\sigma_1})) \;\; \Rightarrow \;\; \min_{\sigma_1} (-\log \sigma_1) \tag{2}$$

Eq. 2 suggests that we simply need to maximize the predicted standard deviation, or equivalently the entropy of the predicted Gaussian. Thus we call it a *max-entropy regularizer* $\Omega$. It can be easily used in any existing neural network architecture:

$$\min_{\theta} \; -\log(\sigma(h|\theta)) \tag{3}$$

where $\sigma(h|\theta)$ denotes the predicted standard deviation of hidden unit $h$ given the neuron uncertainty prediction parameter $\theta$. For numerical safety, we introduce a margin $b$ in the loss,

$$\min_{\theta} (b - \log(\sigma(h|\theta))^+ \tag{4}$$

This means that the regularization does not punish the model as long as the entropy is larger than a threshold $b$. Note that, this loss design is not absolutely necessary as the increment for $\sigma$ shrinks (since the gradient is $-\frac{1}{\sigma}$) during training, and thus never reaches infinity with a finite number of updates. However, one can think of it as an early-stopping mechanism for this regularization term.

So far we have introduced the stochasticity to the smallest unit of a network, i.e., a single neuron. To compute the value of the regularizer over a mini-batch consisting of $N$ training samples, we need to aggregate the entropy of multiple neurons and set the margin $b$ on the aggregation. How to aggregate exactly is task-dependent, and we next provide some suggestions for three tasks including network pruning, adversarial defense, and label noise defense.

**Pruning**  For network pruning, we aggregate entropy over samples for each neuron, and then penalize if that neuron's aggregated entropy is low. To this end, the regularizer is formulated as:

$$\Omega(\theta) = \frac{1}{K} \sum_{j=1}^{K} (b - \frac{1}{N} \sum_{i=1}^{N} \log(\sigma_{i,j}))^+, \tag{5}$$

where $i = 1 \ldots N$ indexes the training samples, and $j = 1 \ldots K$ neurons in a layer (i.e., feature channels). This regularizer aims to make neurons very stochastic, to the point of compromising their reliability for computing a supervised learning task. Thus only those neurons that are most useful for the task get their entropy lowered and thus pay the regularization cost. Less useful neurons get their entropy maximized, allowing them to be detected and pruned after training.

**Label Noise**  Different from pruning, we aim to identify uncertain samples. Therefore, we aggregate entropy over neurons for each sample, and prefer high sample-wise entropy, leading to

$$\Omega(\theta) = \frac{1}{N} \sum_{i=1}^{N} (b - \frac{1}{K} \sum_{j=1}^{K} \log(\sigma_{i,j}))^+. \tag{6}$$

With this regularizer, inlier samples pay the entropy cost in order to produce a clean feature for classification to satisfy the supervised learning loss. Outlier samples, caused by either label noise or being out-of-distribution, are anyway hard to classify; the regularizer naturally inflates their entropy since doing so does not impact the supervised learning loss. This high-variance representation in turn reduces their (negative) impact on network fitting.

**Adversarial Defense**  To defend against adversarial samples, we aim to inflate entropy over both the neuron- and sample-axes to produce a highly stochastic model:

$$\Omega(\theta) = \frac{1}{N} \frac{1}{K} \sum_{i=1}^{N} \sum_{j=1}^{K} (b - \log(\sigma_{i,j}))^+ \tag{7}$$

This looks similar to VIB's adversarial defense (Alemi et al., 2017), but with the vital difference that our entropy regularizer is not restricted to the $\mathcal{N}(0, 1)$ prior used in VIB. It can be seen as learning a layer-wise data augmentation policy, which turns out to be very useful for adversarial defense in practice (see Sec. 4.2).

## 3 RELATED WORK

**Connection to VIB and Sparse VD**    Though we derive the max-entropy regularizer from the perspective of a non-informative activation prior, our work is closely related to VIB (Alemi et al., 2017) and sparse variational dropout (VD) (Molchanov et al., 2017a), despite their different perspectives. Specifically, if we replace our infinite-variance Gaussian with a standard Gaussian, it becomes VIB (see Eq. 17 in (Alemi et al., 2017)). The max-entropy regularizer is also linked to Eq. 14 in Sparse VD (Molchanov et al., 2017a), which also encourages large variance/entropy (at a different rate). But again, Sparse VD (Molchanov et al., 2017a) is derived with a completely different motivation: It has an intuitive explanation that the regularizer corresponds to a sparsity prior on the weights. We note that enforcing uncertainty on activations rather than weights has a number of advantages: (i) The weight prior is intractable analytically, which leads to the fact that Sparse VD regularization is itself an approximation. (ii) Deriving the regularizer from a weight prior is unnecessarily complicated for the purpose of sparsifying the model compared to ours. In contrast, our approach sidesteps the need to sample weights and avoids keeping multiple copies of the network, which can potentially improve efficiency (e.g., in memory usage).

Note that stochastic layers have been used in several other works in order to achieve better classification or regression accuracy. Kingma et al. (2015) proposes a generalization of Gaussian dropout where the dropout rates are learned, leading to higher classification accuracy. Natural-parameter networks (NPN) (Wang et al., 2016) is a class of probabilistic neural networks where the input, target output, weights, and neurons can all be modeled by arbitrary exponential-family distributions (e.g., Poisson distributions for word counts) instead of being limited to Gaussian distributions, achieving state-of-the-art performance on classification, regression, and representation learning tasks. To reduce computational cost, Postels et al. (2019) approximates uncertainty estimates using a sampling-free approach and obtains better results on classification and regression tasks.

**Network Compression**    Network compression based on pruning typically uses heuristics based on pruning low-importance weights or low-activation neurons (Molchanov et al., 2017b; Wen et al., 2016), often assisted by sparsity-enhancing priors such as lasso (Wen et al., 2016). We avoid the complication of Bayesian learning of weights by proposing a simpler and direct activation prior that predisposes neurons towards deactivation unless necessary to solve the supervised task.

**Adversarial Defense**    Our method is related to existing randomization-based methods for adversarial defense (Liu et al., 2018; Xie et al., 2018; Alemi et al., 2017; Ye & Zhu, 2018; Liu et al., 2019). However unlike these studies which use a fixed distribution for noise (Liu et al., 2018; Alemi et al., 2017), a learned model distribution for effective randomization (Liu et al., 2019), image perturbations (Xie et al., 2018) or a learned adversarial data-generating distribution (Ye & Zhu, 2018), our randomization-based defense is both *learned*, and *data-dependent* since the variance at each layer is generated based on the output of the previous layer.

**Label Noise Robustness**    A number of existing label noise-robust deep learning approaches require a subset of noisy data to be reliably re-annotated (cleaned) to verify which samples contain noise (Lee et al., 2017; Jiang et al., 2018). In contrast, some others including SE-SNN do not rely on additional human noise annotation. These methods address label noise by either iterative label correction via bootstrapping (Reed et al., 2015), adding additional layers on top of a softmax classification layer to estimate the noise pattern (Sukhbaatar et al., 2015; Goldberger & Ben-Reuven, 2017), or loss correlation (Patrini et al., 2017). By allocating large uncertainty to outlying samples, SE-SNN can handle both label noise and out-of-distribution samples with correct labels. This approach to label noise robustness is appealingly simple in that it requires neither explicit detection of noisy samples, nor additional annotation. However label noise robustness has been largely ignored by existing SNNs.

## 4 EXPERIMENTS

Experiments are carried out to evaluate the efficacy of the proposed framework in three applications: neural network pruning, adversarial attack defense and learning with label noise.

### 4.1 NEURAL NETWORK PRUNING

**Competitors** We follow the architecture/dataset combinations used in most recent neural network pruning studies, including LeNet-5-Caffe[1] network on MNIST (LeCun, 1998), VGG-16 (Simonyan & Zisserman, 2015) on CIFAR10 (Krizhevsky & Hinton, 2009) and a variant of VGG–16 on CIFAR100. Under these settings, the proposed method is compared with the following contemporary state-of-the-art methods including Generalized Dropout (GD) (Srinivas & Babu, 2016), Group Lasso (GL) (Wen et al., 2016), Sparse Variational Dropout (VD) (Molchanov et al., 2017a), Structured Bayesian Pruning (SBP) (Neklyudov et al., 2017), Bayesian Compression with Group Normal Jeffreys Prior (BC-GNJ) and Group Horseshoe Prior (BC-GHS) (Louizos et al., 2017), Sparse l0 Regularization (L0) and L0 with separate $\lambda$ for each layer (L0-sep) (Louizos et al., 2018), Variational Information Bottleneck (VIBNet) (Dai et al., 2018), and Network Slimming (NS) (Liu et al., 2017).

**Evaluation Metrics** Following the majority of the existing evaluations, we monitor the test error while focusing on the following three compression/complexity metrics: (a) Model size: The ratio of nonzero weights in the compressed network versus the original model. (b) FLOPs: The number of floating point operations required to predict a label from an input image during test[2]. (c) Run-time memory footprint: The ratio of the space for storing hidden feature maps during run-time in the pruned network versus the original network.

**Training** While many existing studies remove redundant *weights*, we remove redundant *neurons* during compression. Therefore, we can use the pipeline proposed in Molchanov et al. (2017b). Specifically, after training the neural network, an initial batch pruning stage is followed by a loop of removing the least important neuron and fine-tuning. Since SE-SNN is designed to discount unimportant neurons through inflating their *pre-activation* variance, we find that the network achieves this by simultaneously assigning high-variance and negative mean. As a result, a large portion of redundant neurons never activate their RELU non-linearity. In the initial batch pruning stage, neuron inactivity thus provides a single-step pruning criterion before the iterative pruning begins (and one that is guaranteed not to affect the test accuracy since these neurons propagate no information). The pruning then enters the second stage where the least important neuron removal + fine-tuning loop continues until reaching the target trade-off between accuracy and model compression objectives. Since one neuron/channel is removed at each iteration, the accuracy never drops sharply. We set the uncertainty loss/regularizer weighting factor and margin $b$ (Eq. 5) as 0.0001 and 4, respectively.

**Testing** During testing, only the mean of the learned distribution is passed between layers. So there are no additional parameters and inference cost compared to the network's deterministic counterpart. The distribution generation branches are only used during training to identify redundant neurons.

**Results on MNIST** The most commonly used benchmark and architecture is MNIST with LeNet-5-Caffe. We follow the standard training and testing protocols. The results are shown in Table 1. It is clear that SE-SNN achieves the best performance on FLOPs, run-time memory footprint, and test error. In terms of model size, our model is only comparable with the state of art. This is because it does not prune the linear layer as much as other methods. Instead, it focuses on pruning the convolutional filters, hence the excellent performance on FLOPS and memory footprint.

**Results on CIFAR10** For CIFAR10, several VGG16 variants and training protocols have been proposed in different works. We use the standard VGG16 architecture but change the dimension of linear layers from 4096 to 512, as in (Louizos et al., 2017; Dai et al., 2018). The results in Table 2 compare our method with (Louizos et al., 2017; Dai et al., 2018). The error rate for VIBNet in parentheses was obtained by further fine-tuning the pruned architecture. Our model achieves the best performance across all the evaluation metrics and error rates.

---

[1]https://github.com/BVLC/caffe/tree/master/examples/mnist

[2]Following the setting in (Dai et al., 2018), we count each multiplication as a single FLOP and exclude additions, which is also consistent with most prior work

| Methods | Model size (%) | FLOPs (Mil) | Memory (%) | Error (%) |
|---|---|---|---|---|
| GD (Srinivas & Babu, 2016) | 1.38 | 0.250 | 32.00 | 1.1 |
| GL (Wen et al., 2016) | 23.69 | 0.201 | 19.35 | 1.0 |
| VD (Molchanov et al., 2017a) | 9.29 | 0.660 | 60.78 | 1.0 |
| SBP (Neklyudov et al., 2017) | 19.66 | 0.213 | 21.15 | **0.9** |
| BC-GNJ (Louizos et al., 2017) | 0.95 | 0.283 | 35.03 | 1.0 |
| BC-GHS (Louizos et al., 2017) | **0.64** | 0.153 | 22.80 | 1.0 |
| L0 (Louizos et al., 2018) | 8.92 | 1.113 | 85.82 | **0.9** |
| L0-sep (Louizos et al., 2018) | 1.08 | 0.389 | 40.36 | 1.0 |
| VIBNet (Dai et al., 2018) | 0.83 | 0.094 | 15.55 | 1.0 |
| SE-SNN | 2.35 | **0.061** | **11.08** | **0.9** |

Table 1: Compression results on MNIST using LeNet-5-Caffe.

| Methods | Model size (%) | FLOPs (Mil) | Memory (%) | Error (%) |
|---|---|---|---|---|
| BC-GNJ (Louizos et al., 2017) | 6.57 | 141.50 | 81.68 | 8.6 |
| BC-GHS (Louizos et al., 2017) | 5.40 | 121.90 | 74.82 | 9.0 |
| VIBNet (Dai et al., 2018) | 5.30 | 70.63 | 49.57 | 8.8(8.5) |
| SE-SNN | **2.57** | **53.61** | **49.41** | **8.0** |

Table 2: Compression results on CIFAR10 using VGG16.

**Results on CIFAR100**    We compare with the study in (Liu et al., 2017), which uses the same VGG16 variant that replaces two fully connected layers with three convolutional layers. This architecture improves accuracy at the expense of FLOPs and memory. The results in Table 3 show that our model produces the best compression result while maintaining comparable accuracy. Note that the 26.2% error rate achieved by our model is identical to that of the original network. So if the accuracy drop is used as a compression metric, our model is as good as any competitor.

## 4.2 ADVERSARIAL DEFENSE

**Experimental Setting**    Following Alemi et al. (2017), we focus on two types of adversarial attacks: Fast Gradient Sign (FGS) (Goodfellow et al., 2015) and an optimization-based attack – CW-L2 (Carlini & Wagner, 2017). We evaluate untargeted FGS attacks with attack magnitude $\epsilon$ ranging from 0.0 to 0.5 and untargeted CW-L2 attack on models trained on MNIST. We use a popular architecture including three FC layers with 1024, 1024 and 256 output neurons respectively. The third FC layer is implemented as a stochastic layer. We set the uncertainty loss/regularizer weighting factor and margin as 0.1 and 16, respectively. Results averaged over 20 runs are reported. We compare against the original (undefended) network, termed as 'Baseline', Deep VIB (Alemi et al., 2017) using the variational information bottleneck, Bayesian Adversarial Learning (BAL) (Ye & Zhu, 2018) putting a distribution on the adversarial data-generating process and Adv-BNN (Liu et al., 2019) learning a BNN to incorporate the effective randomness and using adversarial training to seek the best model distribution. Since our model is stochastic, we follow the best practice recommended in (Athalye et al., 2018) for attacking stochastic models: We compute the expected gradient over multiple stochastic samples for each input when constructing attacks. This is because using the expected gradient over multiple posterior samples produces a better gradient estimator for the attacker, allowing it to generate samples that are much harder to defend against (Athalye et al., 2018). For the FGS attack, we evaluate two settings, namely normal training and adversarial training, the latter of which generates and uses FGS attack samples during training to increase adversarial robustness.

**Results**    From the comparative results shown in Figure 2a, we can see that our SE-SNN outperforms the competing defense methods over a range of FGS attack strengths when trained with adversarial attack samples (Figure 2a(R)) or with only normal samples (Figure 2a(L)). The advantage of SE-SNN is particularly pronounced when the attack magnitude is large. We can also see from Figure 2b that under the stronger CW attacks the Baseline now fails completely. In this case SE-SNN provides the most effective defense. It is worth pointing out that, unlike BAL (Ye & Zhu, 2018), which learns their models with explicit adversarial sampling, our SE-SNN can also work with training with only normal samples - Figure 2a suggests that our SE-SNN beats BAL even without being trained with adversarial attack samples (comparing SE-SNN in Figure 2a(L) to BAL in Figure 2a(R)).

| Methods | Model size (%) | FLOPs (Mil) | Memory (%) | Error (%) |
|---|---|---|---|---|
| NS-Single (Liu et al., 2017) | 24.90 | 250.50 | - | 26.5 |
| NS-Best (Liu et al., 2017) | 20.80 | 214.80 | - | 26.0 |
| VIBNet (Dai et al., 2018) | 15.08 | 203.10 | 73.80 | 25.9(**25.7**) |
| SE-SNN | **14.93** | **181.31** | **70.16** | 26.2 |

Table 3: Compression results on CIFAR100 using a VGG16 variant.

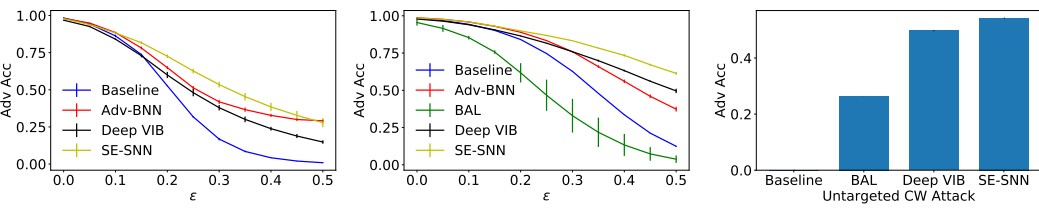

(a) Untargeted FGS (L: normal training, R: adversarial training)     (b) Untargeted CW-L2

Figure 2: Adversarial defense accuracy (mean + standard deviation) under untargeted FGS and CW attacks on MNIST.

## 4.3 ROBUSTNESS AGAINST LABEL NOISE

In this section, tasks including digit recognition and person re-identification (ReID) are considered. Compared to Digit recognition, ReID is much more challenging. Particularly, ReID is an instance recognition task, which aims to match people under disjoint camera views. Although the performance of state-of-the-art ReID models on public benchmarks approaches saturation, ReID with label noise remains an unsolved and under-studied problem.

**Datasets**    The label noise robustness of our SE-SNN is evaluated for digit recognition using MNIST and ReID using Market-1501 (Zheng et al., 2015) benchmark. Market1501 is collected by 6 cameras and contains 751 training identities (12,936 images) and 750 test identities (19,281 images). The test set is organized into a query set and a gallery set. Cumulative Matching Characteristics (CMC) ranks and mean Average Precision are used as the performance measure.

**Noise Generation**    We consider patterned label noise which is more common in practice. Specifically, for MNIST, one class label will be flipped to a different one at a strength $p$. The flipping pattern is fixed for each run but varies across different runs following (Hendrycks et al., 2018). For Market-1501, we use the commonly used ResNet-50 (He et al., 2016) trained on the clean data to obtain the feature of each training sample and find the most visually similar samples using feature Euclidean distance. Then for randomly selected training samples, their identity labels are assigned to that of the most similar sample that has a different identity, imitating human annotation errors.

**Competitors**    *Baseline:* The main baseline is the original network without adding our stochastic layers. For a fair comparison, we add a parallel layer to the penultimate (feature) layer in the baseline to guarantee the two networks have the same number of parameters. The input of the added layer is the same as the feature layer, and its output is element-wise added along with the output of the feature layer to get the final feature, which is then fed to a fully connected layer for computing the classification loss. *Bootstrap_hard* and *Bootstrap_soft* (Reed et al., 2015): Both iteratively use the model-predicted labels to refine the original labels that are potentially corrupted by noise. They differ in whether the updated label is binary or continuous. *Forward Correction* (Patrini et al., 2017): This model predicts the label corruption matrix (capturing the label flipping pattern) by first training a classifier on the noisy labels followed by corruption matrix estimation using the resulting softmax probabilities. The estimated matrix is then employed to regularize the retraining of the model. For MNIST, we stick to the original setting, which uses the $argmax$ at the 97th percentile of softmax probabilities for label noise detection. For the ReID experiment, we replace this with the $argmax$ over all softmax probabilities for a given class following what Patrini et al. (2017) did on datasets of more classes such as CIFAR100. *CleanNet* (Lee et al., 2017): different from other models, this one requires a subset of noisy training samples to be re-annotated by a more reliable source (i.e., cleaned). It then learns the similarity between class and query-embedding vectors, which is further used to detect noisy samples for sample pruning. Having a cleaned subset gives CleanNet

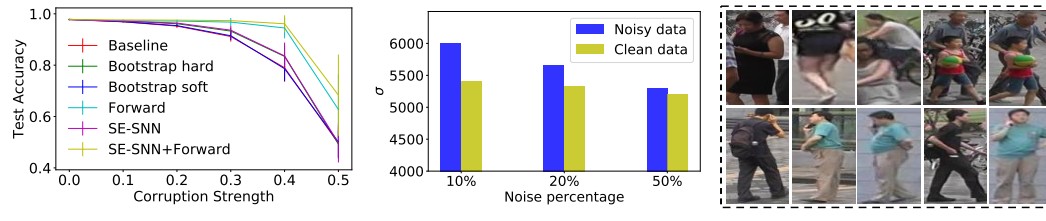

(a) Results on MNIST    (b) Variance comparison on Market    (c) Samples with extreme variance

Figure 3: Label noise results on MINIST (a) and ablation study results on Market-1501 (b,c). In (c), the top row shows samples with the highest variance and bottom row those with the lowest when there is no label noise. The high variance uncertain samples correspond to poor detections and occlusions.

| Strengths | 10% | | | | 20% | | | | 50% | | | |
|---|---|---|---|---|---|---|---|---|---|---|---|---|
| Models | mAP | Rank1 | Rank5 | Rank10 | mAP | Rank1 | Rank5 | Rank10 | mAP | Rank1 | Rank5 | Rank10 |
| B | 25.87 | 51.46 | 70.21 | 76.81 | 23.49 | 48.44 | 67.85 | 74.67 | 20.74 | 44.04 | 64.32 | 72.27 |
| H | 25.76 | 51.08 | 70.11 | 77.06 | 23.40 | 48.25 | 67.34 | 74.40 | 19.87 | 42.87 | 63.37 | 71.40 |
| S | 25.50 | 50.47 | 69.43 | 76.32 | 24.08 | 49.51 | 68.77 | 75.62 | 20.72 | 43.86 | 64.49 | 72.55 |
| C | 26.64 | 52.47 | 70.90 | **77.41** | 24.28 | 49.54 | 68.35 | 75.33 | 19.90 | 43.01 | 63.13 | 71.34 |
| F | 21.17 | 45.88 | 64.79 | 71.69 | 18.91 | 42.11 | 62.12 | 69.99 | 15.34 | 36.06 | 56.23 | 64.55 |
| SE-SNN | **27.30** | **53.21** | **71.03** | 77.38 | **24.72** | **49.87** | **68.90** | **76.10** | **21.32** | **44.86** | **65.03** | **72.93** |

Table 4: Re-ID results on Market-1501 with label-noise. Model abbreviations: **B**: ResNet-Baseline, **H**: Bootstrap_hard (Reed et al., 2015), **S**: Bootstrap_soft (Reed et al., 2015), **C**: CleanNet (Lee et al., 2017), **F**: Forward Correction (Patrini et al., 2017).

an unfair advantage over other compared models. The model requires at least 5 FC layers so cannot be implemented for the MNIST backbone. For person ReID, 10% of the training set without noise is used as a clean reference set to train CleanNet using the author-provided code. After training, 20% of the whole training set deemed most likely to be noisy are removed before the final ReID model is trained on the remainder. Note that unlike the four competitors, our SE-SNN does not have any additional steps to refine the label or prune the training samples. As explained earlier, it works by discounting/neutralizing the negative influence of noisy samples (hence also works on out-of-distribution samples with correct labels). Having said that, it can be easily combined with a label refinement procedure such as the corruption matrix based one in Forward Correction (Patrini et al., 2017).

**Results**    The results on MNIST are shown in Figure 3a. We can see that even without any label correction/refinement procedure, our SE-SNN is already very competitive – it is only beaten by Forward Correction when the corruption strength (percentage of samples with label noise) is large (>0.2). Once our SE-SNN is combined with the same procedure adopted by Forward Correction (SE-SNN+Forward), we achieve the best result. On the more challenging ReID task in Market-1501 (Table 4), our SE-SNN also achieves the best overall performance - even beating CleanNet which needs an additional cleaned training subset. Note that the strongest competitor on MNIST, Forward Correction now is the weakest. This is because, as admitted in (Patrini et al., 2017), it struggles with estimating an accurate correction matrix given a small number of training samples per class (e.g., 500 per class in CIFAR100). The < 20 class size in Market-1501 means that the label correction procedure only has a detrimental effect. Figure 3b compares the average variance/uncertainty inferred for clean and noisy data in Market-1501. We can see that, indeed the variance of noisy data is larger than that of clean data on average. We also notice that even when there is no label noise, SE-SNN beats the baseline by a clear margin ( 70.20% mAP vs. 67.66%). This suggests that the mechanism of neutralizing outlying samples is still in play when the labels are clean. To validate this, we show some examples of outliers (those with the largest variance) in Figure 3c when there is no label noise. It is clear that these are mostly caused by either poor person detection or occlusion (i.e., outliers). In contrast, images with the smallest variance mostly contain people detected perfectly and without occlusion – the model is thus most confident about the feature representations for these.

More SE-SNN implementation details for the label noise robust learning and more results can be found in the Appendix A. Further analyses across all tasks including hyperparameter analysis, solving multiple tasks in one model, effects of the number of stochastic layers are given in Appendix B.

## 5 CONCLUSION

We proposed a simple and effective stochastic neural network framework. Our model is related to VIB and variational dropout, but provides a simpler and more direct realization via neuron regularization by a non-informative activation prior. Our extensive experiments show that this simple framework has diverse benefits for network pruning, adversarial defense and label noise robust learning.

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

| Strengths | 10% | | | | 20% | | | | 50% | | | |
|---|---|---|---|---|---|---|---|---|---|---|---|---|
| Models | mAP | Rank1 | Rank5 | Rank10 | mAP | Rank1 | Rank5 | Rank10 | mAP | Rank1 | Rank5 | Rank10 |
| B | 0.41 | 0.83 | 0.75 | 0.98 | 0.56 | 0.82 | 0.74 | 0.62 | 1.25 | 2.04 | 1.48 | 1.56 |
| H | 0.60 | 0.76 | 1.34 | 1.12 | 0.70 | 0.97 | 0.57 | 0.77 | 0.93 | 1.82 | 1.55 | 0.97 |
| S | 0.78 | 0.93 | 0.85 | 0.95 | 0.82 | 1.48 | 1.45 | 1.38 | 0.51 | 1.11 | 0.85 | 0.61 |
| C | 0.83 | 1.45 | 0.98 | 0.90 | 0.44 | 0.53 | 0.96 | 0.58 | 1.08 | 1.68 | 1.85 | 1.66 |
| F | 1.03 | 1.64 | 1.98 | 1.53 | 0.82 | 1.50 | 1.12 | 0.86 | 0.56 | 1.48 | 1.63 | 1.97 |
| SE-SNN | 0.43 | 1.11 | 0.88 | 0.74 | 0.24 | 0.68 | 0.67 | 0.44 | 0.94 | 1.46 | 1.42 | 1.11 |

Table A.1: Standard deviation value of Re-ID results on Market-1501 with label-noise.

Naftali Tishby, Fernando C Pereira, and William Bialek. The information bottleneck method. In *Allerton*, 1999.

Hao Wang, SHI Xingjian, and Dit-Yan Yeung. Natural-parameter networks: A class of probabilistic neural networks. In *NIPS*, 2016.

Wei Wen, Chunpeng Wu, Yandan Wang, Yiran Chen, and Hai Li. Learning structured sparsity in deep neural networks. In *NIPS*, 2016.

Cihang Xie, Jianyu Wang, Zhishuai Zhang, Zhou Ren, and Alan L. Yuille. Mitigating adversarial effects through randomization. In *ICLR*, 2018.

Nanyang Ye and Zhanxing Zhu. Bayesian adversarial learning. In *NIPS*, 2018.

Zhonghui You, Jinmian Ye, Kunming Li, and Ping Wang. Adversarial noise layer: Regularize neural network by adding noise. *CoRR*, abs/1805.08000, 2018.

Liang Zheng, Liyue Shen, Lu Tian, Shengjin Wang, Jingdong Wang, and Qi Tian. Scalable person re-identification: A benchmark. In *ICCV*, 2015.

# A   EXPERIMENTAL DETAILS OF LABEL NOISE

For the experiments on MNIST, we follow (Patrini et al., 2017) to train a neural network with two FC layers of dimension 128, where both FC layers are implemented as stochastic layers. For Market-1501, following most recent state-of-the-art ReID models we use ResNet-50 (He et al., 2016) as the backbone. In this model, only the last unit of the final residual block is implemented as a stochastic layer.

For Market-1501, we train our model for two stages: (1) We fine-tune ImageNet pretrained ResNet-50 (for identity classification) on Market-1501 training set for $60,000$ iterations. We use batch size 32 and Adam optimizer (Kingma & Ba, 2015) with learning rate $3.5 \times 10^{-4}$. (2) We retain the model parameters trained in stage (1) and extend the last unit to a stochastic layer with randomly initialized $\sigma$ layer. Then, we train the stochastic layer and the classifier for another $20,000$ iterations with a lower learning rate $5 \times 10^{-4}$. The margin $b$ is set as 4 in the max-entropy regularization loss (Eq. 6) and a loss weight $10^{-2}$ is assigned.

For both benchmarks, due to the randomness of noisy samples in selection and label reassignment, multiple runs of experiments are conducted. The standard deviation values of Re-ID results on Market-1501 with label-noise are shown in Table A.1.

# B   FURTHER ANALYSIS

## B.1   HYPER-PARAMETERS ANALYSIS

In this section, we analyze the impact of loss weight, which is denoted as $\omega$, and the margin $b$ for regularizers of all tasks on MNIST. We report the FLOPs of the compressed model (Network pruning), adversarial defense accuracy under the FGSM attack (Adversarial defense) and test accuracy of a model trained with 40% label noise (Label noise) in Figure B.1.

In Figure B.1(top), we evaluate the effects of having different values of $\omega$ from $[10^{-1}, 10^{-2}, 10^{-3}]$. We can see that, for all three values and across the three tasks, the performance of our method is only marginally impacted by $\omega$ and the overall best performance is achieved when $\omega = 10^{-2}$.

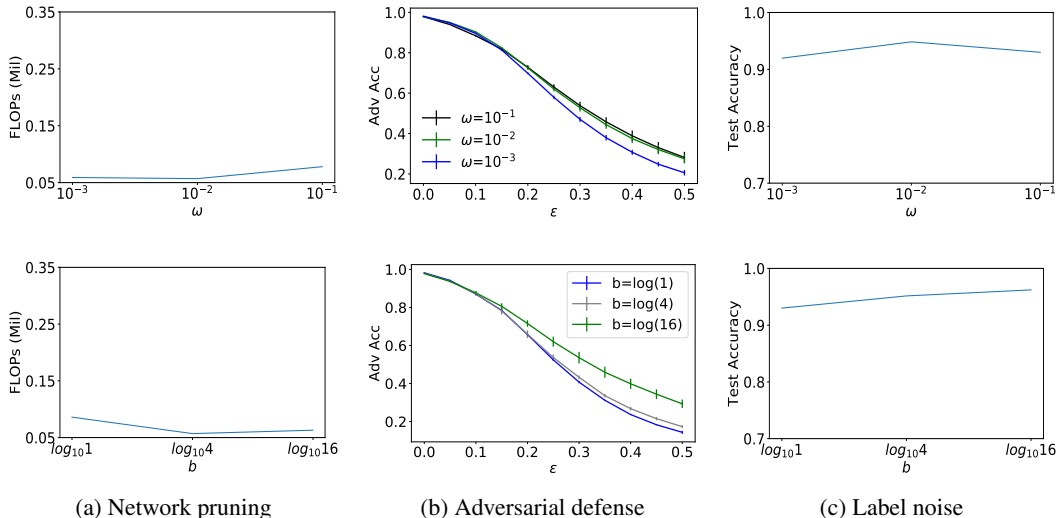

|                    |                        |                  |
| ------------------ | ---------------------- | ---------------- |
| (a) Network pruning | (b) Adversarial defense | (c) Label noise |

Figure B.1: Impact analysis of loss weight $\omega$ (top) and margin $b$ (bottom) on MNIST.

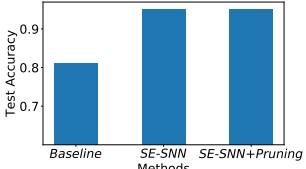

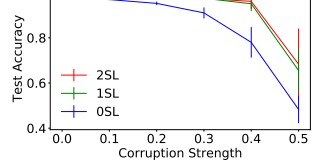

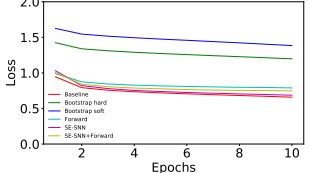

Figure B.2: Test accuracy of different methods trained on MNIST with label noise.

Figure B.3: Effect of the number of stochastic layers on MNIST with label noise.

Figure B.4: Convergence Curve on MNIST with label noise .

In Figure B.1(bottom), we investigate the sensitivity of our model to the values of margin $b$. Three values, $[\log(1), \log(4), \log(16)]$ are considered. Recall that our SE-SNN uses a non-informative prior to regularize the uncertainty of neuron activations, and a margin $b$ is employed to upper-bound the standard deviation of the stochastic layers in SE-SNN. Therefore, larger values of $b$ will lead to a higher degree of uncertainty of stochastic layers. From Figure B.1(bottom), we can see that increasing the value of $b$ improves the performance of our method on adversarial defense and label noise. However, for network pruning, smaller $b$ is preferred. Overall, our model is insensitive to the value of $b$.

## B.2 COMBINATION OF NETWORK PRUNING AND LABEL NOISE

The effectiveness of our SE-SNN has been demonstrated on the three tasks separately, but there is no reason why it cannot be deployed to deal with multiple tasks simultaneously. To validate this, in this experiment, we train SE-SNN on MNIST with 40% label noise while conducting model pruning. We use the same architecture with three FC layers (784-128-128-10) as the previous label noise experiments, and make the first two layers stochastic layers. We compare the trained model with two competitors, including the baseline model (without stochastic layers) and our normal SE-SNN (without pruning) both trained with label noise.

From the results in Figure B.2, we can see that we get a pruned SE-SNN (145-26-15-10) trained with label noise achieving 95.20% test accuracy, which is even slightly better than our normal SE-SNN (95.17%) and much better than the baseline (81.04%).

## B.3 EFFECT OF THE NUMBER OF STOCHASTIC LAYERS

In this section, we study the effect of the number of used stochastic layers (SLs). The experiments are run on the label noise robust learning task with network comprising two FC layers. For the network

with one SL, we make the penultimate layer stochastic. Note that the network with zero SL is the baseline. Figure B.3 shows the test accuracy of models with different numbers of SLs used. We can see that increasing the number of SLs helps improve the model's robustness to label noise.

## B.4 CONVERGENCE CURVE

The training efficiency of the proposed method is studied here by examining how fast the algorithm converges. The experiments are run on the label noise robust learning task. Figure B.4 shows the losses of different methods during 10 training epochs. It is observed that Bootstrap and Forward have larger loss values than the rest. This is because they use losses in addition to the classification loss. Overall our method is shown to be as efficient as other methods, despite the fact that we have introduced stochastic layers and used a reparameterization trick during training.

