# OpenReview forum: "Simple and Effective Stochastic Neural Networks"
_ICLR.cc/2020/Conference — Reject_

### Official Review · AnonReviewer3 · 2019-10-21
**Official Blind Review #3**

**Rating:** 3

**Review:**

Summary:
This paper presents an efficient stochastic neural network architecture by directly modeling activation uncertainty and adding a regularization term to encourage high activation variability by maximizing the entropy of stochastic neurons. Compared with other existing approaches, such as Bayesian neural networks and variational information bottleneck, the proposed architecture is simpler to implement and faster to train. The authors also achieve state of the art results in various fields, including network compression by pruning, adversarial defense and learning with label noise.

Major comments:
- Overall, I find the paper is easy to follow and the experimental evaluation shows promising results, but my major concern is about the novelty of this work, given the fact that the structure of the proposed stochastic layers is quite similar to VIBNet.
- The derivation of the max-entropy term is somewhat unclear and I think the paper needs a major revision on this part. The authors suggest using a Gaussian with a finite mean and an infinite variance as the non-informative prior for the produced Gaussian random variable z (in Eq. 1), and then minimize the KL divergence between the produced Gaussian and the infinite-variance Gaussian. However, this may be questionable from a Bayesian viewpoint, in the sense that the infinite variance leads to an improper prior as the variance increases without bound and thus may produce an improper posterior distribution. This is not discussed and needs to be clarified in Sect. 2 (Max-entropy Regularization).
- There are things unclear in the derivation (last line of Eq. 2), since log(\sigma_2) trends to infinity.
- In addition, the penalty terms for different types of tasks are directly given only based on some of the assumptions that the authors have made, there does not seem to be any theoretical justification for such choices.

Minor comments:
- Some of the notations used in this paper seem a bit confusing, which may hinder readability. For example, on page 3, in “The non-informative prior is a Gaussian with arbitrary mean (\mu_1) and infinite variance (\sigma_1)”, I guess \sigma means the standard deviation? I would like to recommend using N(\mu, \sigma^2) to denote a Gaussian distribution, where \sigma means the standard deviation and \sigma^2 the variance.
- On page 3, in “where \sigma(h|\theta) denotes the predicted standard deviation of hidden unit h given the neuron uncertainty prediction parameter \theta”, there is no discussion on the neuron uncertainty prediction parameter. Does it means the predicted standard deviation is again parameterized by \theta?

**Experience Assessment:**

I have read many papers in this area.

**Review Assessment: Checking Correctness Of Derivations And Theory:**

I assessed the sensibility of the derivations and theory.

**Review Assessment: Checking Correctness Of Experiments:**

I assessed the sensibility of the experiments.

**Review Assessment: Thoroughness In Paper Reading:**

I read the paper at least twice and used my best judgement in assessing the paper.

---

> ### Author Response · Authors · 2019-11-12
> **Response to Reviewer #3**
>
> We thank the reviewer for the comments. We have revised the paper accordingly and would like to clarify several things:
>
> Q1. Novelty:
> Though the structure of our proposed method is similar with DeepVIB [Alemi et al., 2017], the motivation is very different. In DeepVIB, there is an intuitive explanation that the regularizer corresponds to a sparsity prior on the weights. In contrast, the regularizer introduced in our model enforces uncertainty on activations rather than weights. Besides, our key contribution is to introduce a non-informative prior instead of using the standard Gaussian as an informative prior. For the three applications of network compression by pruning, adversarial defense and learning with label noise, the standard Gaussian is not the best prior, as we have empirically observed in this paper:  tuning the covariance matrix usually leads to better performance. On the contrary, using our non-informative prior stably produces better performance without extra tunings.
>
>
> Q2. Infinite-variance Gaussian
> Indeed, the proposed informative prior is improper, and is designed to be so. Importantly, it does not necessarily mean the posterior distribution is also improper in practice, as this depends on the exact process of optimisation. As we mentioned in page 3, Eq (4). The gradient is -1/\simga, which means that the increment for gradient shrinks when \sigma increases. So it never reaches infinity with a finite number of training updates. Furthermore, we have a margin loss the stops increasing \sigma when it reaches a certain finite value.
>
>
> Q3. Eq2
> It is clear from Eq 2 that \sigma2 is irrelevant to the final objective, so it can be disregarded for whatever value it is.
>
>
> Q4. Theoretical Justification
> We can think of margin loss as an early-stopping mechanism – it stops pushing the entropy larger when it surpasses a certain threshold. For different problems, we design different aggregation methods (across sample and/or feature axis) which are intuitively motivated by the specific problem settings. We have explained those deigning choices in page 3 for pruning, label noise, and adversarial defense respectively.
>
> Minor:
> We have revised the paper to differentiate variance and standard deviation. Predicted standard deviation is parameterized by \theta as Figure 1 shows.

---

### Official Review · AnonReviewer1 · 2019-10-22
**Official Blind Review #1**

**Rating:** 3

**Review:**

This paper proposed SE-SNN, a type of stochastic neural networks that maximize the entropy in stochastic neurons along with the prediction accuracy. The authors argue that maximizing the entropy operates as a form of regularization to force the entropy into the least significant neurons, and that Increasing the diversity/randomness results in more robust models. Experiments show that SE-SNN outperform several baselines tasks such as network pruning, adversarial defense, and learning with noisy lables.

Several closely related references are missing. The idea of producing distributions in each layer (i.e., using stochastic layers) is not new and is closely related to the work on local reparameterization trick and variational dropout [1] (predecessor of the cited sparse variational dropout), and various works that directly model neurons as distribution [2, 3].

Built on top of the reparameterization trick, the idea of maximizing the entropy of neurons to regularize the network is interesting. Such regularization is somewhat similar to sparsity regularization on neurons, which forces the information to concentrate on a small portion of the neurons.

The numbers for different methods in Table 1 are very close to each other. Without standard deviation it is difficult to evaluate the performance.

Note that in Table 1 the best accuracy is actually achieved by SBP and L0. However, in Table 2 and 3 (experiments on CIFAR-10 and CIFAR-100), these two baselines are missing. Is there a reason why these baselines are excluded?

Also, looking at the results from Louizos et al., 2018, which is the most recent baseline among those chosen by the authors, they actually use WideResNet rather than VGG. Comparing to VGG, WRN performs significantly better. For example, the compressed network by Louizos et al., 2018 achieves an error rate of 3.83% on CIFAR-10, versus 8% from SE-SNN pruned VGG.

For the experiments on adversarial defense, the two attacks used seem rather weak (FGS from 2015 and CW-L2 from 2017). It may be the state-of-the-art attack around the time of Alemi et al., 2017, which the author claim to follow, but not now.

Minor:

P2: instead -> instead of

[1] Variational Dropout and the Local Reparameterization Trick, NIPS 2015
[2] Natural-Parameter Networks: A Class of Probabilistic Neural Networks, NIPS 2016
[3] Sampling-free Epistemic Uncertainty Estimation Using Approximated Variance Propagation, ICCV 2019

**Experience Assessment:**

I have published in this field for several years.

**Review Assessment: Checking Correctness Of Derivations And Theory:**

I assessed the sensibility of the derivations and theory.

**Review Assessment: Checking Correctness Of Experiments:**

I carefully checked the experiments.

**Review Assessment: Thoroughness In Paper Reading:**

I read the paper thoroughly.

---

> ### Author Response · Authors · 2019-11-15
> **Response to Review#1:**
>
> We thank the reviewer for the comments. We have revised the paper as suggested and would like to clarify several things:
>
> Q1. More related work:
> We have added the suggested work in the related work section.
>
> Q2. Standard Deviation in Table 1 and different baseline in task of MNIST and CIFAR:
> We cropped the results from (Dai et al., 2018), in which they did not show the standard deviation. Note that different papers used different network backbones. For example, the L0 method (Louizos et al., 2018) uses WRN on CIFAR datasets and SBP (Neklyudov et al., 2017) used a variant of VGG. Therefore we consider it to be unfair for direct comparison with them in Table 2.
>
> Q3. VGG rather than WRN:
> We use the same setting as in (Dai et al., 2018), which is the most relevant paper. In (Dai et al., 2018), VGG is also used as the backbone, which has also been commonly used in other works. Indeed, having a stronger backbone would have a big impact on the absolute number of the error rate. However, to show the effectiveness of a model compression algorithm, the backbone has been the same for fair comparison among competitors. Having said that, we have now obtained a new result using WRN-28-10. The error rate is much lower, but the compression rate is also much lower: Our method reached the accuracy of 3.82% while the model size compressed to 81.97% and the FLOPs size decreased to 4.16 billion from 5.24 billion. In comparison, from Figure 4(a) and (b) in (Louizos et al., 2018), the FLOPs is reduced from 350 billion to 325 billion – clearly a different (nonstandard) way to calculate FLOPs is used in (Louizos et al., 2018) because most recent projects report similar FLOPs number as ours (e.g., Park et al., 2018, open source codes such as https://github.com/osmr/imgclsmob/tree/master/pytorch). But the relative compression performance is clearly weaker than ours.
>
> Q4. More attacks:
> We add the latest black-box adversarial attack method NATTACK (Li et al., 2019). The results below show that our SE-SNN is clearly superior to the compared methods.
> +--------------+-------------+--------------+--------------+------------+
> |                  | Baseline | Adv-BNN | Deep VIB | SE-SNN |
> +--------------+-------------+--------------+--------------+------------+
> | NATTACK |    5.94%  |    56.44%  |   82.18%   |  95.00% |
> +--------------+-------------+--------------+--------------+------------+
>
> [1] Jongchan Park, Sanghyun Woo, Joon-Young Lee, and In-So Kweon. BAM: Bottleneck Attention Module. In BMVC 2018.
> [2] Yandong Li, Lijun Li, Liqiang Wang, Tong Zhang, and Boqing Gong. NATTACK: Learning the Distributions of Adversarial Examples for an Improved Black-Box Attack on Deep Neural Networks. In ICML, 2019.

---

### Official Review · AnonReviewer2 · 2019-10-27
**Official Blind Review #2**

**Rating:** 6

**Review:**

This paper presents a simple stochastic neural network, which makes each neuron output Gaussian random variables. The model is trained with reparameterization trick. The authors advocates the adoptation of a non-informative prior, and shows that learning with the prior equals with an entropy-maximization regularization term. The paper presents the design of the regularization term for pruning, learning with label noise, and defensing with adversarial examples. The claims are well supported: the model is indeed simple, and the effectiveness is well supported by experimental results.

I however think *efficiency* of the proposed model needs to be studied as well. Since the proposed algorithm is an approximate inference algorithm via reparametrization trick, it is necessary to see how fast does the approximate algorithm converge. The experiments don't report any convergence curve, or performance under limited time budget. I think there need to be some related results.

Another question is, what is the original inference problem of the designed regularization for pruning, label noise, and adversarial attack, respectively?

Update
====

Thanks for the additional experiments! After reading the rebuttal and other reviewer's comments I decide to keep my score. (Though I am less positive due to the concern of novelty raised by other reviewers.)

**Experience Assessment:**

I do not know much about this area.

**Review Assessment: Checking Correctness Of Derivations And Theory:**

I assessed the sensibility of the derivations and theory.

**Review Assessment: Checking Correctness Of Experiments:**

I assessed the sensibility of the experiments.

**Review Assessment: Thoroughness In Paper Reading:**

I read the paper at least twice and used my best judgement in assessing the paper.

---

> ### Author Response · Authors · 2019-11-12
> **Response to Reviewer #2**
>
> We thank the reviewer for the comments. We have revised the paper according to the suggestions and would like to clarify several things:
>
> Q1. Convergence Curve:
> We have added a convergence curve for label noise setting in Appendix B.4.
>
> Q2. Original Inference Problem:
> We have asked for clarification, and are waiting for responses.

---

### Decision · Program_Chairs · 2019-12-19

**Decision:**

Reject

**Comment:**

This paper proposes to use stacked layers of Gaussian latent variables with a maxent objective function as a regulariser. I agree with the reviewers that there is very little novelty and the experiments are not very convincing.